# Alfalfa, Cabbage, Beet and Fennel Microgreens in Floating Hydroponics—Perspective Nutritious Food?

**DOI:** 10.3390/plants12112098

**Published:** 2023-05-25

**Authors:** Sanja Fabek Uher, Sanja Radman, Nevena Opačić, Mia Dujmović, Božidar Benko, Denis Lagundžija, Valent Mijić, Lucija Prša, Srđan Babac, Jana Šic Žlabur

**Affiliations:** 1Department of Vegetable Crops, University of Zagreb Faculty of Agriculture, Svetošimunska Cesta 25, 10000 Zagreb, Croatia; 2Department of Agricultural Technology, Storage and Transport, University of Zagreb Faculty of Agriculture, Svetošimunska Cesta 25, 10000 Zagreb, Croatia; 3Graduate Studies Horticulture, Organic Agriculture with Agrotourism, University of Zagreb Faculty of Agriculture, Svetošimunska Cesta 25, 10000 Zagreb, Croatia

**Keywords:** floating system, young shouts, nutritional potential, antioxidant capacity, phenolics

## Abstract

Microgreens are young plants of various vegetables, medicinal and aromatic plants, cereals and edible wild plants that were first associated with nouvelle cuisine as decoration in dishes due to their attractive appearance and strong flavor. Recently, they have become more sought after in the market due to their high nutritional value. This is due to the growing interest of consumers in a healthy lifestyle that includes a varied diet with emphasis on fresh, functional foods. Nowadays, commercial production of microgreens is shifting to modern hydroponic systems due to their numerous advantages, such as accelerated plant growth and biomass production, earlier harvesting, and more production cycles that positively affect yield and chemical composition. Therefore, the aim of this study was to determine the content of specialized metabolites and antioxidant capacity of hydroponically grown alfalfa (Medicago sativa) cv. ‘Kangaroo’, yellow beet (Beta vulgaris var. conditiva) cv. ‘Yellow Lady’, red cabbage (Brassica oleracea L. var. rubra) cv. ‘Red Carpet’, and fennel (Foeniculum vulgare) cv. ‘Aganarpo’ microgreens. The highest content of total phenols (408.03 mg GAE/100 g fw), flavonoids (214.47 mg GAE/100 g fw), non-flavonoids (193.56 mg GAE/100 g fw) and ascorbic acid (74.94 mg/100 g fw) was found in fennel microgreens. The highest content of all analyzed chlorophyll pigments (Chl_a 0.536 mg/g fw, Chl_b 0.248 mg/g fw, and TCh 0.785 mg/g fw) was found in alfalfa microgreens. However, in addition to alfalfa, high levels of chlorophyll a (0.528 mg/g fw), total chlorophyll (0.713 mg/g fw) and the highest level of total carotenoids (0.196 mg/g fw) were also detected in fennel microgreens. The results suggest that microgreens grown on perlite in floating hydroponics have high nutritional potential as a functional food important for human health and therefore could be recommended for daily diet.

## 1. Introduction

In recent years, consumers are on the lookout for new food sources rich in health-promoting and disease-preventing bioactive compounds [1,2]. In this context, microgreens have great potential to diversify and improve the human diet due to their attractive morphological characteristics and high nutritional value [3].

Microgreens are young plants of various vegetables, medicinal and aromatic plants, cereals and edible wild plants that are consumed at the stage of fully developed cotyledons and incompletely developed first pair of true leaves. They first appeared in the United States as a product used by high-end restaurants in the creation of upscale dishes (food fashion), but soon this trend spread to all levels of gastronomy [4]. In addition to their decorative value, microgreens soon became known as a very popular healthy food due to their status as a “superfood” or functional food [5]. 

Depending on the species, microgreens contain high concentrations of nutrients such as minerals (P, Mg, Ca, Fe, Zn, Cu), vitamins (C, E, K), and bioactive compounds (phenolics and pigments) that are important for the healthy functioning of the human body. Some species of microgreens contain more nutrients than mature edible plant parts, and their consumption in relatively small amounts can meet daily requirements for certain vitamins and minerals [5,6,7,8,9]. Compared to seeds or mature plants, both sprouts and microgreens are rich in amino acids, fatty acids, and simple sugars as a result of enzymatic degradation of large macromolecules and bioactive compounds [10]. The biochemical changes during seed germination lead to the activation of various enzymes that break down macromolecules into smaller molecules that can be rapidly absorbed by the human organism [11]. 

Microgreens have the potential to increase food security and diversify agricultural production in the world. They are a readily available source of fresh and nutrient-rich food in areas with limited space and water supply because they have much lower requirements for cultivated land, water, and plant nutrients than traditional crops [4,5,12]. Microgreens are ideal for production in urban areas, especially in large vertical planting systems. Cities can thus potentially meet some of their needs for fresh functional foods that do not lose their nutritional and organoleptic properties due to long transportation distances. On the other hand, they become more self-sufficient and less dependent on food supply chains, which is very important in case of natural disasters or the current COVID-19 pandemic, where supply chains are severely disrupted (at risk) [13,14,15,16]. Microgreens production can be sustainable and has a smaller environmental footprint than conventional vegetable production [5].

Although in recent years there are numerous foreign studies [5,17,18,19,20] that address the production factors of microgreens, there are currently no production standards for cultivation of microgreens or guidelines on how to achieve the best nutrient composition of microgreens with the highest possible yield. One of the most important production factors affecting microgreens yield and quality is cultivation technology [5,7,17,21].

The most commonly used substrates for growing microgreens are mixtures of peat substrate, but due to their high price and limited resources, producers are looking for alternative substrates such as sand, coconut and textile fibers, or perlite. There are also concerns about the potential source of pathogenic organisms in organic substrates. Therefore, inert substrates such as perlite are offered as a logical option because they do not support disease development [22,23]. 

In addition to the importance of sustainable agriculture, soilless production is attracting great interest from microgreens producers, as they need to adapt to new market demands. One of the advantages of hydroponic techniques is the more efficient use of water and nutrients. Since the plants are grown under controlled conditions, plant growth is also accelerated [24]. With the choice of the right technique (e.g., ebb and flow) and the proper management of cultivation factors (the nutrient solution can be an appropriate tool to obtain the desired chemical composition of the plant material), it is possible to stimulate the synthesis and accumulation of bioactive compounds and obtain microgreens with high nutritional value [25]. However, before choosing more advanced hydroponic systems, it is important to understand and further research the elementary cultivation systems, such as floating hydroponics to achieve production standards for specified microgreens species.

The aim of this study was to lay the foundations for cultivation of microgreens in floating hydroponics and to investigate the nutritional value and functional potential of alfalfa, red cabbage, yellow beet and fennel microgreens.

## 2. Results

Table 1 shows the yield (kg/m^2^) and total dry matter content (DM, %) of the studied microgreens species. There were significant differences in both traits among the studied species. The obtained yield of microgreens ranged from 0.22 to 2.69 kg/m^2^. The highest yield was obtained with red cabbage microgreens, while the yield of fennel microgreens was 12 times lower. Alfalfa and yellow beet microgreens did not differ statistically in yield (0.85 and 0.72 kg/m^2^, respectively).

Although with the lowest yield, the fennel microgreens in this research were characterized by the highest dry matter content (10.15%). The lowest value of DM (6.25%) was found in red cabbage microgreens and was statistically equal to yellow beet microgreens (6.33%). The average DM in young shoots of the tested species was 7.85%.

The results of specialized metabolites content and antioxidant capacity of the studied microgreens species are presented in Table 2. According to the statistical analysis, there were significant differences (*p* ≤ 0.05) for all analyzed phytochemicals. The ascorbic acid (AsA) content varied from 28.20 to 74.94 mg/100 g fw resulting with an average content of 52.28 mg/100 g fw. The highest AsA content in this research was found in fennel microgreens and the lowest in yellow beet microgreens. Fennel microgreens were characterised by the highest levels of total phenols (TPC), total non-flavonoids (TNFC) and total flavonoids (TFC), with values of 408.03 mg GAE/100 g fw, 193.56 mg GAE/100 g fw and 214.47 mg GAE/100 g fw, respectively. The lowest content of total phenolics (115.31 mg GAE/100 g fw) and total non-flavonoids (51.02 mg GAE/100 g fw) was determined in the red cabbage microgreens. Moreover, the lowest and statistically equal content of total flavonoids was found in red cabbage (64.29 mg GAE/100 g fw) and alfalafa (60.11 mg GAE/100 g fw) microgreens. The antioxidant capacity values of the tested microgreens in this study varied from 2361.18 µmol TE/L (red cabbage) to 2430.01 µmol TE/L (yellow beet), with an average value of 2392.55 µmol TE/L. Alfalfa microgreens (2410.47 µmol TE/L) were among the statistically highest antioxidant capacity, apart from yellow beet microgreens. Also, no justified differences in antioxidant capacity were found between fennel and red cabbage microgreens.

The results of the pigment compounds analyzed: chlorophyll a (Chl_a), chlorophyll b (Chl_b), total chlorophyll (TCh) and total carotenoid (TCa) content are presented in Table 3. Pigment compounds significantly (*p* ≤ 0.05) varied between analyzed microgreens. According to the results of this research, a high value of pigments, TCh and TCa are obtained regardless of the species. The highest content of all analyzed chlorophyll pigments (Chl_a 0.536 mg/g, Chl_b 0.248 mg/g and TCh 0.785 mg/g) was determined in alfalfa microgreens. Fennel microgreens, along with alfalfa microgreens, were characterised by high levels of chlorophyll a (0.528 mg/g) and total chlorophyll (0.713 mg/g), and had the highest content of total carotenoids (0.196 mg/g). Compared to alfalfa, the values of all analyzed pigment compounds in yellow beet microgreens were twice as low (Chl_a 0.249 mg/g, Chl_b 0.108 mg/g, TCh 0.358 mg/g, and TCa 0.089 mg/g).

## 3. Discussion

The yield of hydroponically grown microgreens depends on the plant species, sowing density, hydroponics technique, nutrient solution and substrate used. Achieving satisfactory yield is often a challenge for microgreens growers [5,17,20,26]. According to Murphy et al. [27], the highest yield of beet microgreens can be obtained when seeds are germinated in moist vermiculite and grown using the nutrient film technique. The same authors note that seed treatments prior to sowing can potentially affect beet microgreens yield. 

Peat-based substrates are the main growing media currently used for cultivating microgreens, but they are costly and not reusable [3]. Perlite is a very good growing medium with strong capillary action so that the plants can easily reach the nutrient solution, i.e., nutrients, which is very important because of the very short production cycle that lasts only 10 to 28 days depending on the species [5,7,17,28]. Perlite can also be reused for multiple growing cycles if washed and sterilized after each use, making it suitable for sustainable agriculture [28]. The average yield of microgreens obtained in this study using perlite as substrate was 1.12 kg/m^2^. In the research of Brlek [29], microgreens were grown in three types of substrate. The use of a commercial substrate mixed with perlite gave the highest yield (1.0 kg/m^2^), while cultivation on burlap resulted with the lowest yield (0.6 kg/m^2^). In the aforementioned study, cultivation on a commercial substrate produced the highest yield of beet microgreens. In the study by El-Nakhel et al. [30], parsley (*Petroselinum crispum*) microgreens were grown on a peat-based substrate and harvested 21 days after sowing, which is consistent with the growing period of fennel microgreens in our study, as it is a species of the *Apiaceae* family. Kyriacou et al. [2] state that to achieve high yield of microgreens, additional fertilization after emergence or combined fertilization before sowing and after emergence is required. In addition to the amount and method of application, the form of fertilizer, especially the ratio of ammonium nitrate, can also affect the yield and quality of young shoots. Moderate concentrations of ammonia (15:85; NH_4_^+^:NO_3_^−^) improve plant growth, photosynthesis, chloroplasts, and root structure of Chinese cabbage (*Brassica pekinensis*) compared to pure nitrate (0:100; NH_4_^+^:NO_3_^−^) [2]. In addition, precise formulation and maintenance of nutrient solution composition throughout the growing period can increase the content of specific functional compounds such as glucosinolates in *Brassica* species.

In general, the dry matter content of the raw material depends strongly on several factors, ecological conditions (temperature, humidity, water availability), cultivation method, phenophase of the plant, but also genetic characteristics [5,31,32]. This chemical parameter is also the first indicator of the nutritional quality of the plant material, suggesting that raw material with higher DM values contain a high amount of nutrients, mainly minerals, vitamins, and other phytochemicals. Microgreens species analyzed under this study, generally had low DM values, i.e., high water content (average value of 92%), which was expected due to the plant phenophase (developed cotyledons and incompletely developed first pair of true leaves) and the cultivation method (floating hydroponics). Since, in floating system plants have continuous supply of nutrient solution, respectively water, generally plant material grown under such conditions going to accumulate higher amount of water [33]. The microgreens of yellow beets in this study had a higher DM content compared to the results of Brlek [29] for microgreens of beets in a mixture of substrate and perlite. In view of the above, it must be emphasized that given the high water content, the preservation, storage and manipulation of fresh microgreens present a challenge due to the very short shelf life [7].

Ascorbic acid (AsA) is one of the best-known oxygen scavenging molecules in both plant tissues (protects plants from stress conditions) and human cells having a main role as a powerful antioxidant. Besides a well-known condition of higher synthesis and accumulation rate of AsA when plants are exposed to the stress conditions, the role of plant phenophase has not been studied in such detail. Namely, L-ascorbic acid is crucial agent as the co-factor in the biosynthesis of some plant hormones (ethylene, gibberellic acid, and abscisic acid), whose main role is the regulation of developmental processes as a signal molecule in regulation of flowering time. In mature edible plant parts, reactive oxygen species (ROS) content increases in plant cells since the AsA level strongly decrease affecting also on the photosynthetic apparatus and a decrease in photosynthetic activity [34,35]. Also, different research studies discuss about AsA role in regulation of cell division during plant embryo development, thus proving that the synthesis of AsA increases during leaf growth while declining with the decrease in leaf function as part of the plant senescence [36]. In general, it can be concluded that high values of AsA can be expected in the cotyledon phase (microgreens). Moreover, the photosynthesis rate increases in the cotyledon phase, which also promotes the production of glucose, which is a precursor of AsA synthesis. Based, on the all above, high AsA values recorded in microgreens under this study are expected. Stated is also proven by other studies, Xiao [37] note even six-time higher AsA concentration in microgreens of red cabbage than in red cabbage harvested at the technological maturity stage, furthermore fennel in technological maturity stage on average contains about 32.5 mg/100 g fw of AsA [38], while in this research about two times higher AsA content was recorded in fennel microgreens. It should also be noted that when the AsA content of fennel microgreens was compared with other literature data [39], an approximately 25% higher AsA content was recorded for microgreens analyzed in this research. In the study by Ghoora et al. [39], the AsA content of fennel microgreens grown in substrate enriched with vermicompost was 60.2 mg/100 g fw. Giordano et al. [40] found that microgreens of 4 species from the *Apiaceae* family grown on peat-based substrate and irrigated with ¼ strength Hoagland nutrient solution had AsA levels ranging from 71.6 (caraway) to 150 (dill) mg AA /100 g fw. The AsA content of yellow beet microgreens in this study is consistent with the results of Xiao et al. [9] for red beet microgreens (28.8 mg/100 g fw), while Brazaitytė et al. [41] reported values ranging from 0.69 to 7.49 mg/g in beet microgreens grown under supplemental UV-A irradiation.

Polyphenolics as the main compounds of plant secondary metabolism are responsible for protecting plant cells from free radicals and maintaining the balance of active oxygen metabolism by inhibiting reactive oxygen species. The mentioned function is the most pronounced but also closely related to the role of polyphenolic compounds in plant growth and development as a main protection antioxidants [42,43]. Cotyledon phase is a vital and first moment of plant response to the light in the process of photomorphogenesis, while the generation of chloroplast formation as photosynthetically functional organelles and greening (chlorophyll synthesis) are crucial for further plant response to light stress. Polyphenolic compounds, apart from their role in the response of plants to various types of stresses (drought, salinity, high and low temperatures, etc.) are also key to the response of plants to light stress, thus protecting them from excessive and harmful UV-B irradiation [43,44]. Based on the results of this research and the above-mentioned role of phenolics in plant development phases, higher TPC, TFC and TNFC content is expected in the cotyledon phase (microgreens) compared to the later phenophase. This can be supported by comparing the total phenol content of for example fennel microgreens with the fennel bulbs in full maturity stage. The authors Salama et al. [38] reported an average value of TPC content in fennel bulb of about 5.5 mg/g dw, while in this research TPC content of fennel microgreens was 7 times higher value. By comparing results of this research with other literature data [39], specifically TPC, TFC and TNFC content of fennel microgreens, can be noted that in this research several times higher values were obtained highlighting this type of plant material as a very valuable source of important antioxidants. According to Brazaitytė et al. [41], the use of higher UV-A LED irradiance can increase the total phenolic content in beet microgreens to 1.28 mg/g. Hernández-Adasme et al. [45] also investigated the total phenolic content of beet microgreens under different light treatments. Depending on the treatment, the content ranged from 9.58 to 12.48 mg GAE /g fw. Seleh et al. [46] studied chard microgreens grown on different growing media (consisting of different ratios of vermicast, sawdust, perlite, PittMoss and Pro-mix) and reported an average total phenolic content of 151.44 mg GAE/g. Altuner et al. [47] studied alfalfa microgreens grown on a mixture of peat, cocopite, and perlite and reported a total phenolic content of 327.89 mg GAE/100 g dw, while the total flavonoid content was 366.27 mg QE/100 g dw. The total flavonoid content in fennel microgreens from the study by Ghoora et al. [39] was 5.5 mg/100 g fw which is 39 times less than the value from our study.

The key moment in the development of the first cotyledons is the transition from the phase of skotomorphogenesis to photomorphogenesis, which is mainly controlled by light. In the cotyledon stage of development, this is manifested by the expansion and greening of the cotyledons and the formation of chloroplasts, i.e., the photosynthetic apparatus [43]. Obviously, the cotyledon stage is when the plant first develops mechanisms to respond to light and begins to accumulate chlorophyll. This suggests that microgreens will be a rich source of chlorophyll. In addition to the plant organism, pigments such as chlorophyll and carotenoids are also extremely important for the human organism. They are powerful antioxidants and have a synergistic effects with other bioactive compounds, which is why they have numerous beneficial effects on human health [48,49,50]. Altuner et al. [47] report values determined for total chlorophyll, chlorophyll a, chlorophyll b and carotenoids in alfalfa microgreens to be as follows: 26.62, 19.44, 7.18, and 5.23 μg/g TA fw. In a 2020 study by Ghoora et al. [39], the total chlorophyll content of fennel microgreens was 90.3, while the values for chlorophyll a and chlorophyll b were 56.1 and 34.4 mg/100 g fw, respectively. According to Giordano et al. [40], *Apiaceae* species microgreens contain an average of 1.04 mg/g weight of total chlorophyll. Wieth et al. [20] tested four commercial substrates and three concentrations of nutrient solutions for total chlorophyll and total carotenoid content of red cabbage microgreens. The values of total chlorophyll averaged 0.410 mg/g fw while the average content of total carotenoids was 0.008 mg/g fw. In the study by Kowitcharoen et al. [11] red cabbage microgreens were cultivated in soilless culture using Kinocloth substrate and the results for total chlorophyl and carotenoid content were 39.79 and 12.08 mg/100 g, respectively.

Considering the high content of bioactive compounds found in all microgreens studied, high values of antioxidant capacity of these samples were expected. Polyphenols, vitamins, pigments and other phytochemicals, due to their specific molecular structure, have significant potential to inhibit free radicals, maintain the balance of the oxidation process in plant cells and protect them from excessive accumulation of ROS radicals, oxidative stress and death. From the group of specialized metabolites, flavonoids are one of the most important compounds related to antioxidant capacity. In this study, fennel microgreens are the richest in flavonoids, clearly standing out from other species and possessing high antioxidant capacity. In general, all antioxidant capacity values of the studied microgreens are high, proving that these samples have high nutritional and functional potential. As confirmed by other studies, microgreens have high antioxidant activity and thus great potential as valuable health foods [39,51,52,53]. The total antioxidant activity of alfalfa microgreens is 990.417 mg TE /g DW according to Altuner et al. [47]. Hernández-Adasme et al. [45] reported an average value of antioxidant capacity of beet microgreens of 37.87 mg TE/g fw, while Ghoora et al. [39] put the antioxidant activity of fennel microgreens at 22.8 µmol TE/g. 

## 4. Materials and Methods

### 4.1. Plant Material

The experiment was conducted in the spring growing season of 2021 in a heated greenhouse at the Experimental Station of the Department of Vegetable Crops at the University of Zagreb Faculty of Agriculture in Croatia. Microgreens of alfalfa (*Medicago sativa*) cv. ‘Kangaroo’, yellow beet (*Beta vulgaris* var. *conditiva*) cv. ‘Yellow Lady’, red cabbage (*Brassica oleracea* L. var. *rubra*) cv. ‘Red Carpet’ and fennel (*Foeniculum vulgare*) cv. ‘Aganarpo’ (Remsprout Company—Rem S.R.L.; https://www.remsprout.com/retail-shop, accessed on 10 May 2023) were cultivated using floating hydroponics technique. Sowing was carried out on April 30 in polystyrene containers (0.5 × 0.3 m) filled with perlite with seed consumption of 300 (alfalfa), 357 (beet), 267 (cabbage), and 333 (fennel) g/m^2^. Seeds were covered with perlite, irrigated with water, and containers were placed in a greenhouse with optimal temperature for germination of cultivated species (temperature: 20 °C and relative humidity: 60%). During germination, the growing media was manually watered and covered with black polyethylene film until germination was completed. Germination of red cabbage and alfalfa was observed 3 days after sowing, and that of yellow beet and fennel 5 days later. After germination, the containers were placed in a floating hydroponics basin (1 × 2 m) filled with Tesi nutrient solution used for hydroponic cultivation of leafy vegetables and aromatic and medicinal plants [54,55,56,57,58]. The following salts in mg/L were used to prepare the nutrient solution (NS): KNO_3_—784.5, KH_2_PO_4_—272.2, K_2_SO_4_—20.9, Ca(NO_3_)_2_ × 4H_2_O—972.5, MgSO_4_ × 7H_2_O—246.3, NH_4_NO_3_—28.0, FeEDTA 13%—16.8, H_3_BO_3_—1.86, CuSO_4_ × 5H_2_O—0.19, MnSO_4_ × 4H_2_O—0.85, ZnSO_4_ × 7H_2_O—1.15, and Na_2_MoO_4_ × 2H_2_O—0.12 (EC 2.3 mS/cm, pH 5.5–5.8).

Harvesting was done manually with scissors on 19 May, 20 days after sowing (red cabbage and alfalfa microgreens) and on 25 May, 25 days after sowing (yellow beet and fennel microgreens). After yield measurement, a sample of 100 g of each species was prepared for laboratory analysis, which was carried out at the Department of Agricultural Technology, Storage and Transport at University of Zagreb Faculty of Agriculture.

### 4.2. Abiotic Parameters of the Air and Nutrient Solution

A multiparameter meter (Hanna instruments HI98194, Romania) was used to regularly measure the hydrogen potential, known as pH, and the electrical conductivity, known as EC (mS/cm) values of the solution during the microgreens cultivation. The pH of the NS averaged 6.43 and did not deviate much from the average with a maximum deviation of 6.97 and a minimum deviation of 6.05. The EC value averaged 2.38 mS/cm and did not vary much with a maximum deviation of 2.62 mS/cm and a minimum deviation of 2.03 mS/cm.

In addition, air temperature and relative humidity were measured in the minimum and maximum values in the greenhouse using a table thermohygrometer (Agrologistika d.o.o., Čakovec, Croatia) from 4 to 25 May. Air temperature values ranged from 12 to 34 °C. The values of air relative humidity in the greenhouse ranged from 21.6 to 79.8%.

### 4.3. Determination of Total Dry Matter

Total dry matter content (DM, %) was determined by drying to constant mass at 105 °C using a standard laboratory protocol according to AOAC [59]. Total dry matter content was expressed as a percentage using Equation (1):(1)$$\mathrm{DM}\;\left(\%\right)=\frac{m_{2}-m_{0}}{m_{1}-m_{0}}\times 100$$
where *m*_0_ (g) is the mass of glassware; *m*_1_ (g) is the mass of glassware with the microgreens samples before drying; *m*_2_ (g) is the mass of glassware with the microgreens samples after drying.

### 4.4. Determination of Specialized Metabolites Content

The following specialized metabolites were determined in microgreens: ascorbic acid content (AsA), total phenols (TPC), total flavonoids (TFC), total nonflavonoids (TNFC) such as phenolic acids, tannins, stilbenes, lignans and pigment compounds (total chlorophylls, chlorophyll a, chlorophyll b, total carotenoids).

The AsA content was determined by titration with 2,6-dichloroindophenol (DCPIP) according to the standard laboratory method available in AOAC [59]. AsA was isolated from the fresh microgreens with 2% (*v*/*v*) oxalic acid; first 10 g ± 0.01 of fresh plant material was weighed and homogenized with 100 mL of 2% (*v*/*v*) oxalic acid. Homogenized solution was filtered through Whatman filter paper, the filtrate was obtained in a volume of 10 mL and titrated with freshly prepared DCPIP till the appearance of a characteristic pink coloration. The final AsA content was calculated according to Equation (2) and expressed as mg/100 g fresh weight.
(2)$$\mathrm{AsA}\;(\mathrm{mg}/100\;g\;\mathrm{fw})=\frac{V\times F}{D}\times 100$$
where is: V is the volume (mL) of DCPIP; F is the factor of DCPIP, 0.1336245 for alfalfa and red cabbage and 0,1589189 for yellow beets and fennel; D is the sample mass (g) used for titration.

The content of total phenolics (*TPC*), including flavonoids (*TFC*) and non-flavonoids (*TNFC*), was determined spectrophotometrically (Shimadzu, UV 1650 PC), using a method based on a color reaction of phenols with the Folin-Ciocalteu reagent, measured at 750 nm according to Ough and Amerine [60]. Ethanol solution (80%, *v/v*) and reflux were combined for the isolation of polyphenolic compounds from microgreens. Fresh plant material of 10 g ± 0.01 was weighed into an Erlenmeyer flask and the first 40 mL of 80% EtOH (*v/v*) was added. The prepared sample was heated to boiling point and additionally refluxed for 10 min. After 10 min, the sample was filtered through Whatman filter paper into a 100 mL volumetric flask. After filtration, the remainder of the sample was transferred to the Erlenmeyer flask and another 50 mL of 80% EtOH (*v/v*) was added while the procedure was repeated under reflux for 10 min. The sample was filtered into the same volumetric flask, the filtrates were combined, and the flask was made up to the mark with 80% EtOH (*v/v*). The extract thus prepared was used for the reaction with the Folin-Ciocalteu reagent. To a volumetric flask of 50 mL, 0.5 mL of the ethanolic extract and the following chemicals were added: 30 mL of distilled water (dH_2_O), 2.5 mL of the prepared Folin-Ciocalteu reagent (1:2 with dH_2_O), and 7.5 mL of saturated sodium carbonate solution (Na_2_CO_3_); the flask was filled to the mark with dH_2_O, and the prepared sample was allowed to stand at room temperature for 2 h with constant shaking. The same ethanolic extracts prepared for TPC determination were also used for TNFC content determination, while TNFC separation was performed according to the following procedure: 10 mL of the ethanolic extract was added to a 25 mL volumetric flask, then 5 mL of 37% HCl (1:4, *v/v*) and 5 mL of formaldehyde were added. The prepared samples were aerated with nitrogen (N_2_) and left at room temperature for 24 h in a dark place. After 24 h, the same Folin-Ciocalteu reaction was performed as for TPC. The absorbance of blue color in both TPC and TNFC reactions was measured spectrophotometrically at 750 nm using distilled water as blank. Gallic acid was used as external standard and the final concentration of TPC, TFC and TNFC content was expressed as mg GAE/100 g fresh weight. The difference between total phenols and non-flavonoids shows the amount of flavonoids.

Pigment compounds were determined according to the method described by Holm [61] and Wettstein [62]. For the determination of pigments, approximately 0.2 g of microgreens were weighed and mixed with a total volume of 15 mL of acetone (p.a.) in three steps. The solutions were homogenized using a laboratory homogenizer (IKA, UltraTurax T-18, Staufencity, Germany), then filtered and measured using acetone (p.a.) as a blank. Total chlorophylls (TCh), chlorophyll a (Chl_a), chlorophyll b (Chl_b) and total carotenoids (TCa) were determined and absorbance values were measured spectrophotometrically (Shimadzu, 1900i, Kyoto, Japan) at wavelengths of 662, 644 and 440 nm. The pigment content was calculated by including measured absorbance values in the Holm–Wettstein Equations (3)–(6) and the results were expressed in mg/g.
Chl_a = 9.784 × A662 − 0.990 × A644 (mg/L)(3)
Chl_b = 21.426 × A644 − 4.65 × A662 (mg/L)(4)
TCh = 5.134 × A662 + 20.436 × A644 (mg/L)(5)
TCa = 4.695 × A440 − 0.268 × TCh (mg/L)(6)

### 4.5. Determination of Antioxidant Capacity

Antioxidant capacity was determined using the ABTS assay, which is one of the most commonly used methods for estimating antioxidant capacity and was described by Miller et al. [63]. The 2,2′-azinobis (3-ethylbenzothiazoline-6-sulfonic acid) and potassium persulfate from Sigma-Aldrich were used. To prepare a stable ABTS+ solution, 88 μL of a 140 mmol K_2_S_2_O_8_ solution was mixed with 5 mL of ABTS solution and left for 12–16 h at room temperature in the dark. On the day of analysis, a 1% ABTS solution was prepared in 96% ethanol and the absorbance was measured at 734 nm. Directly in a cuvette, 160 μL of the extract was mixed with 2 mL of a 1% ABTS-+ solution, and after 5 min, the absorbance was measured spectrophotometrically (Shimadzu, 1900i, Kyoto, Japan). 96% ethanol was used as the blank. Trolox (6-hydroxy-2,5,7,8-tetramethylchroman-2-carboxylic acid, Sigma-Aldrich, St. Louis, MO, USA) was used as an antioxidant standard. Final results were calculated based on a calibration curve and expressed in μmol TE/L (μmol Trolox per liter).

### 4.6. Statistical Analysis

The experiment was laid out according to a randomized complete block design with three replications, where one treatment in each repetition was represented by 3 polystyrene containers. All laboratory analyzes were performed in triplicate and the obtained results were statistically analyzed using PROC GLM in SAS software system, version 14.3 [64]. One-way analysis of variance (ANOVA) was used. Means were compared using t-test (LSD) and considered significantly different at *p* ≤ 0.05. In addition to the results, different letters are given in the tables to indicate significant statistical differences between the different microgreens at *p* ≤ 0.05 and standard deviation (SD) is also indicated. 

## 5. Conclusions

The activities carried out in this research provide additional data on the production of microgreens on perlite in floating hydroponics as a perspective nutritious food, which is confirmed by its high antioxidant capacity. Red cabbage microgreens had the highest yield but also the lowest dry matter content. Fennel microgreens were characterized by the highest content of dry matter, ascorbic acid, phenols, flavonoids and non-flavonoids, as well as high content of carotenoids and chlorophylls. In addition, the highest content of total carotenoids, chlorophyll a and chlorophyll b, and total chlorophyll was found in alfalfa microgreens. Careful selection of substrate, nutrient solution, and cultivation technique can fit well with sustainable, environmentally friendly agriculture, which is the main advantage over conventional farming. The information from this study is a good basis for further research on more innovative hydroponic techniques for growing microgreens such as ebb and flow, aquaponics and aeroponics with different substrates (vermiculite, coconut fiber) and nutrient solutions (biofortification) that can further improve the nutritional value of microgreens. Further detailed analysis of bioactive compounds found in hydroponically grown microgreens should also be performed.

## Figures and Tables

**Table 1 plants-12-02098-t001:** Yield and total dry matter content of microgreens.

Microgreens	Yield (kg/m^2^)	DM (%)
Alfalfa	0.85 ^b^ ± 0.08	8.68 ^b^ ± 0.06
Red cabbage	2.69 ^a^ ± 0.09	6.25 ^c^ ± 0.15
Yellow beet	0.72 ^b^ ± 0.13	6.33 ^c^ ± 0.08
Fennel	0.22 ^c^ ± 0.08	10.15 ^a^ ± 0.16
Average	1.12	7.85
LSD	0.1801	0.2228

DM—dry matter. Results are expressed as mean ± standard deviation. Different letters show significant statistical difference between mean values with *p* ≤ 0.05.

**Table 2 plants-12-02098-t002:** Specialized metabolites content and antioxidant capacity of microgreens.

Microgreens	AsA (mg/100 g fw)	TPC (mg GAE/100 g fw)	TNFC (mg GAE/100 g fw)	TFC (mg GAE/100 g fw)	Ant_Cap (µmol TE/L)
Alfalfa	56.75 ^b^ ± 2.56	120.16 ^c^ ± 3.73	60.06 ^b^ ± 0.49	60.11 ^c^ ± 4.20	2410.47 ^ab^ ± 26.69
Red cabbage	49.24 ^c^ ± 2.55	115.31 ^d^ ± 1.89	51.02 ^d^ ± 0.40	64.29 ^c^ ± 2.24	2361.18 ^c^ ± 33.28
Yellow beet	28.20 ^d^ ± 2.23	137.90 ^b^ ± 0.70	54.14 ^c^ ± 0.21	83.76 ^b^ ± 0.77	2430.01 ^a^ ± 9.46
Fennel	74.94 ^a^ ± 2.68	408.03 ^a^ ± 1.29	193.56 ^a^ ± 1.98	214.47 ^a^ ± 2.88	2368.54 ^bc^ ± 10.19
Average	52.28	195.35	89.70	105.56	2392.55
LSD	4.7276	4.171	1.9647	5.2886	42.239

AsA—ascorbic acid; TPC—total phenol content; TNFC—total non-flavonoid content; TFC—total flavonoid content; Ant_cap—antioxidant capacity. Results are expressed as mean ± standard deviation. Different letters show significant statistical difference between mean values with *p* ≤ 0.05.

**Table 3 plants-12-02098-t003:** Pigment compounds of microgreens.

Microgreens	Chl_a (mg/g)	Chl_b (mg/g)	TCh (mg/g)	TCa (mg/g)
Alfalfa	0.536 ^a^ ± 0.06	0.248 ^a^ ± 0.02	0.785 ^a^ ± 0.07	0.174 ^a^ ± 0.02
Red cabbage	0.370 ^b^ ± 0.06	0.204 ^ab^ ± 0.04	0.573 ^b^ ± 0.10	0.122 ^b^ ± 0.02
Yellow beet	0.249 ^c^ ± 0.07	0.108 ^c^ ± 0.02	0.358 ^c^ ± 0.08	0.089 ^c^ ± 0.02
Fennel	0.528 ^a^ ± 0.02	0.184 ^b^ ± 0.02	0.713 ^a^ ± 0.04	0.196 ^a^ ± 0.01
Average	0.421	0.186	0.607	0.145
LSD	0.1019	0.0478	0.1431	0.0371

Chl_a—chlorophyll a content; Chl_b—chlorophyll b content; TCh—total chlorophyll content; TCa—total carotenoid content. Results are expressed as mean ± standard deviation. Different letters show significant statistical difference between mean values with *p* ≤ 0.05.

## Data Availability

All research data from the experiment are available from the authors.

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
