# Peer review of "Alfalfa, Cabbage, Beet and Fennel Microgreens in Floating Hydroponics—Perspective Nutritious Food?"

_plants, 2023, doi:10.3390/plants12112098_

Round 1
Reviewer 1 Report
The manuscript entitled „Case study of microgreens in floating hydroponics – perspective nutritious food” I propose to accept the Plants for publication in the journal after taking into account minor comments by the authors. The manuscript was written with great care and accuracy. Unfortunately, it presents results only in four microgreens and the experiment was performer without using any new analytical technique. The results are very general. Very simple ststistical analyzes were also performer. Furthermore, it is nowhere explained which compounds we refer to as TNFC. I have a tot of doubts about the very preparation of the extract for the analysis of the content of bioactive compounds. ” Erlenmeyer flask and the first 40 mL of 80% EtOH (v/v) was added. The prepared sample 324 was heated to boiling point and additionally refluxed for 10 min” it is quite controversial.
Lines 25-28 : please copmplete the information about the form of the material in which determined former chlorophylls and carotenoids.
Line 261-263: where the research material was obtained - seeds
Lines 286-287: please explain abbreviations
Line 295: what device was used
Line 305: TNFC – what kind of compounds are examples
The results should not be repeated and this is at work especially with bioactive compounds and pigments, necessarily to change.
Line 166: „Brassica” italic
Line 308: what kind of method?
Line 324: Erlenmeyer flask? Without reflux condenser?
Line 339: HCl what concentration
Line 365: 2,20-azinobis…?
Line 367: pleas change „mM” into „mmol”
Whay TFC is converted to CTH a are calculated from the difference in TPC and TNFC which in turn are converted to gallic acid eqivalent?
Reviewer 2 Report
The nutritional and health-promoting quality of microgreens largely depends on the growing conditions used. The presented manuscript focuses on the basic analysis of the content of dry matter, phenolic compounds, lipophilic pigments and on the determination of the antioxidant potential of 4 types of microgreens. The results presented in the manuscript have been subjected to statistical analysis and are properly commented in the results section. The discussion of the results focuses mainly on the description of the changes taking place during the formation of microgreens. However, there is no broader discussion of the results with data for microgreens grown in other conditions, e.g. type of substrate, fertilization, lighting. Such data is available for funnel, red cabbage or alfalfa. Please emphasize the innovative nature of the research, both in the Introduction and in the Discussion.
The manuscript needs to be completed and corrected as listed below:
Lines 98, 122 and 138 - "Results are expressed as ± standard deviation" replace "Results are expressed as mean ± standard deviation"
Line 132 - wrong value, should be 0.713 instead of 0.716
Table 3 - the content of chlorophyll and carotenoid pigments may be given in mg/100g fw, as well as the content of phenolic compounds.
Line 308 - instead of "2,6-dichlorindophenol" it should be 2,6-dichloroindophenol
Line 316 - the given notation suggests that the result was divided and not multiplied by 100 (converted to 100 g)
Line 317 - please specify the DCIPP factor in the equation
Line 346 - enter the full name for the abbreviation CTH. Please justify the purpose for which catechol was used, since the content of flavonoids was calculated from the difference in the determination of the content of phenolic compounds by the Folin-Ciocalteu method before and after flavonoids precipitation with formaldehyde.
Line 374 - enter the full name for the abbreviation TE.
Reviewer 3 Report
Abstract/title and methods
Microgreens of alfalfa, red cabbage, yellow beet, and fennel - missing/not included either in Title or in keywords, should be included
Species and varieties, are included in materials and method, but should be also included in abstract.
Analytical methods - the major problem of the work.
All the analysis of the different microgreens is not including a minimum characterisation of compounds (a minimum HPLC analysis).
The colorimetric assays are inappropriate for quantification of compounds. There are plenty and hundreds of publications about these microgreens and different species/varieties to be able to check for composition and a minimum number of compounds/phytochemicals to look for to incorporate the data in the manuscript (by a minimum HPLC analysis, already available in the majority of labs doing research in food chemistry - either in their own institution, either in collaborations, or technical support from other units).
Otherwise the information can not be positively evaluated in 2023.
Round 2
Reviewer 2 Report
I recommend accepting the manuscript with the additions and corrections made.
Author Response
Thank you for your valuable time and feedback.
Best regards,
Sanja Radman
Corresponding author
Reviewer 3 Report
Authors incorporated corrections and additional data of composition are not possible to be included, therefore, the request can not be answered anyway.
Author Response
Thank you for your valuable time and suggestions. We have made some changes in English language and style and marked them in yellow in the manuscript.
Best regards,
Sanja Radman
Corresponding author